# BATCH REINFORCEMENT LEARNING THROUGH CONTINUATION METHOD

**Yijie Guo**[1]  **Shengyu Feng**[1]  **Nicolas Le Roux**[2]  **Ed Chi**[2]  **Honglak Lee**[1,2]  **Minmin Chen**[2]
[1]University of Michigan      [2]Google AI
{guoyijie,shengyuf}@umich.edu  {nlr,edchi,honglak,minminc}@google.com

## ABSTRACT

Many real-world applications of reinforcement learning (RL) require the agent to learn from a fixed set of trajectories, without collecting new interactions. Policy optimization under this setting is extremely challenging as: 1) the geometry of the objective function is hard to optimize efficiently; 2) the shift of data distributions causes high noise in the value estimation. In this work, we propose a simple yet effective policy iteration approach to batch RL using global optimization techniques known as continuation. By constraining the difference between the learned policy and the behavior policy that generates the fixed trajectories, and continuously relaxing the constraint, our method 1) helps the agent escape local optima; 2) reduces the error in policy evaluation in the optimization procedure. We present results on a variety of control tasks, game environments and a recommendation task to empirically demonstrate the efficacy of our proposed method.

## 1 INTRODUCTION

While RL is fundamentally an *online* learning paradigm, many practical applications of RL algorithms, e.g., recommender systems [5, 7] or autonomous driving [36], fall under the *batch* RL setup. Under this setting, the agent is asked to learn its policy from a fixed set of interactions collected by a different (and possibly unknown) policy commonly referred to as the behavior policy, without the flexibility to gather new interactions. Realizing the interactive nature of online RL has been hindering its wider adoptions, researchers strive to bring these techniques offline [24, 11, 20, 23, 31, 12, 21, 2, 32, 8]. We focus on policy optimization under batch RL setup. As pointed out in [3, 26], even with access to the exact gradient, the loss surface of the objective function maximizing the expected return is difficult to optimize, leading to slow convergence. Chen et al. [8] show that the objective function of expected return exhibits sub-optimal plateaus and exponentially many local optima in the worst case. Batch setup makes the learning even harder as it adds large variance to the gradient estimate, especially when the learned policy differs from the behavior policy used to generate the fixed trajectories. Recent works propose to constrain the size of the policy update [27, 28] or the distance between the learned policy and the behavior policy [14, 21]. The strength of that constraint is a critical hyperparameter that can be hard to tune [28], as a loose constraint does not alleviate the distribution shift while a strict one results in conservative updates.

Here we propose to address the challenges using continuation methods [35, 6, 17]. Continuation methods attempt to solve the global optimization problem by progressively solving a sequence of new objectives that can be optimized more efficiently and then trace back the solutions to the original one. We change the objective function of policy optimization by including an additional term penalizing the KL divergence between the parameterized policy $\pi_\theta$ and the behavior policy. We then gradually decrease the weight of that penalty, eventually converging to optimizing the expected return. With this additional constraint, we benefit from more accurate policy evaluation in the early stage of training as the target policy is constrained to be close to the behavior policy. As training continues, we relax the constraint and allow for more aggressive improvement over the behavior policy as long as the policy evaluation is still stable and relatively reliable, i.e. with a small enough variance. By doing so, the proposed method exhaustively exploits the information in the collected trajectories while avoiding the overestimation of state-action pairs that lack support.

The contributions of this paper are as follows: (1) We propose a soft policy iteration approach to batch RL through the continuation method. (2) We theoretically verify that in the tabular setting with exact gradients, maximizing KL regularized expected return leads to faster convergence than optimizing the expected return alone. Also, our method converges to the globally optimal policy if there are sufficient data samples for accurate value estimation. (3) We demonstrate the effectiveness of our method in reducing errors in value estimation using visualization; (4) We empirically verify the advantages of our method over existing batch RL methods on various complex tasks.

## 2 RELATED WORK

**Batch Reinforcement Learning.** Off-policy reinforcement learning has been extensively studied [11, 20, 30, 23, 31], with many works [12, 21, 2] focusing on variants of Q-learning. Fujimoto et al. [12], Kumar et al. [21] investigated the extrapolation error in batch RL resulting from the mismatch of state-action visitation distribution between the fixed dataset and the current policy, and proposed to address it by constraining the action distribution of the current policy from deviating much from the training dataset distribution. Recent works [29, 33] studied policy iteration under batch RL. The Q function is estimated in the policy evaluation step without special treatment while the policy updates are regularized to remain close to the prior policy with a *fixed* constraint. To further reduce uncertainty in Q learning, an ensemble of Q networks [21, 29] and distributional Q-function [2, 33] are introduced for the value estimation. [34, 18] use the KL divergence between the the target policy and the behavior policy as a regularization term in the policy update and/or value estimation. The constraint is controlled by a fixed weight of the KL regularization or a fixed threshold for the KL divergence. While all of these works apply a *fixed* constraint determined by a sensitive hyper-parameter to control the distance between the behavior/prior policy and the target policy, we focus on *gradually relaxed* constraints.

**Constrained Policy Updates.** Several works [27, 1, 15] studied constrained policy updates in online settings. Kakade & Langford [19] show that large policy updates can be destructive, and propose a conservative policy iteration algorithm to find an approximately optimal policy. Schulman et al. [27] constrain the KL divergence between the old policy and new policy to guarantee policy improvement in each update. Grau-Moya et al. [15] force the policy to stay close to a learned prior distribution over actions, deriving a mutual-information regularization between state and action. Cheng et al. [9] propose to regularize in the function space. Again these methods focused on a *fixed* constraint while we are interested in continuing relaxing the constraint to maximize the expected return eventually. Also none of these methods have been extensively tested for batch RL with fixed training data.

**Continuation Method.** Continuation method [35] is a global optimization technique. The main idea is to transform a nonlinear and highly non-convex objective function to a series of smoother and easier to optimize objective functions. The optimization procedure is successively applied to the new functions that are progressively more complex and closer to the original non-context problem, to trace their solutions back to the original objective function. Chapelle et al. [6] use the continuation method to optimize the objective function of semi-supervised SVMs and reach lower test error compared with algorithms directly minimizing the original objective. Hale et al. [17] apply the continuation method to l1-regularized problems and demonstrate better performance for compressed sensing problems. Inspired by prior works, we employ the continuation method to transform the objective of batch RL problems by adding regularization. We gradually decrease the regularization weight to trace the solution back to the original problem.

## 3 METHOD

In classical RL, an agent interacts with the environment while updating its policy. At each step $t$, the agent observes a state $s_t \in \mathcal{S}$, selects an action $a_t \in \mathcal{A}$ according to its policy to receive a reward $r_t = r(s_t, a_t) : \mathcal{S} \times \mathcal{A} \to \mathbb{R}$ and transitions to the next state $s_{t+1} \sim \mathcal{P}(\cdot|s_t, a_t)$. The state value of a policy $\pi$ at a state $s$ is $V^\pi(s) = \mathbb{E}_{s_0=s, a_t \sim \pi(\cdot|s_t), s_{t+1} \sim \mathcal{P}(\cdot|s_t, a_t)} [\sum_{t=0}^{\infty} \gamma^t r(s_t, a_t)]$. $\gamma \in [0, 1]$ is the discounting factor. At each step, the agent updates the policy $\pi$ so that the expected return $V^\pi(\rho) = \mathbb{E}_{s \sim \rho}[V^\pi(s)]$ (where $\rho$ is the initial state distribution) is maximized.

In batch RL, the agent is not allowed to interact with the environment during policy learning. Instead it has access to a fixed set of trajectories sampled from the environment according to a behavior policy[1]. A trajectory $\{(s_0, a_0, r_0), (s_1, a_1, r_1), \cdots, (s_T, a_T, r_T)\}$ is generated by sampling $s_0$ from the initial state distribution $\rho$, sampling the action $a_t \sim \beta(\cdot|s_t)$ at the state $s_t$ and moving to $s_{t+1} \sim \mathcal{P}(\cdot|s_t, a_t)$ for each step $t \in [0, 1, \cdots, T]$. The length $T$ can vary among trajectories. We then convert the generated trajectories to a dataset $\mathcal{D} = \{(s_i, a_i, r_i, s_i')\}_{i=1}^{N}$, where $s_i'$ is the next state after $s_i$ in a trajectory.

The goal of batch RL is to learn a parameterized policy $\pi_\theta$ with the provided dataset to maximize the expected return $V^\pi(\rho)$. In Sec. 3.1, we will first introduce a new objective function $\tilde{V}^{\pi,\tau}(\rho)$, i.e. the expected return of policy $\pi$ with KL regularization term and the regularization weight $\tau$. With exact gradients, $\tilde{V}^{\pi,\tau}(\rho)$ can be optimized more efficiently than the original objective $V^\pi(\rho)$. With the

---

[1]If the behavior policy is not known in advance, it can be fitted from the data [30, 7].

continuation method, solving a sequence of optimization problems for $\tilde{V}^{\pi,\tau}(\rho)$ with decaying value of $\tau$ converges toward optimizing $V^{\pi}(\rho)$ and makes the optimization easier. In Sec. 3.2, we derive soft policy iteration with KL regularization to optimize $\tilde{V}^{\pi,\tau}(\rho)$, without the assumption of exact gradients. Finally, in Sec. 3.3, we propose a practical batch RL algorithm with value estimation for target policy based on this theory.

### 3.1 OPTIMIZING EXPECTED RETURN WITH KL REGULARIZATION

In batch RL, the distribution of the trajectories generated by the behavior policy can be very different from that of the learned policy. We thus restrict the learned policy to stay close to the behavior policy via the regularization of KL divergence. Define the soft state value of a policy $\pi$ at a state $s$ as

$$\tilde{V}^{\pi,\tau}(s) = \mathbb{E}_{s_0=s, a_t \sim \pi(\cdot|s_t), s_{t+1} \sim \mathcal{P}(\cdot|s_t, a_t)} \left[ \sum_{t=0}^{\infty} \gamma^t \left( r(s_t, a_t) - \tau \log \frac{\pi(a_t|s_t)}{\beta(a_t|s_t)} \right) \right], \quad (1)$$

where the temperature parameter $\tau$ controls the deviation from $\beta$. The new objective function becomes $\tilde{V}^{\pi,\tau}(\rho) = \mathbb{E}_{s \sim \rho}[\tilde{V}^{\pi,\tau}(s)]$. This KL regularized objective differs from the original objective $V^{\pi}(\rho)$, which however can be recovered as $\tau \to 0$.

As pointed out in [3], even with exact gradients, the objective function $V^{\pi}(\rho)$ is still difficult to optimize due to its highly non-smooth landscape. Mei et al. [26] further prove that, in a tabular setting with softmax parameterized policy and exact gradients, the vanilla policy gradient method (i.e. directly updating the parameters of policy $\pi$ to maximize $V^{\pi}(\rho)$ with gradient descent) converges to the global optimal policy at a convergence rate $\mathcal{O}(1/t)$, while the entropy-regularized policy gradient enjoys a significantly faster linear convergence rate $O(e^{-t})$. Motivated by this line of work, we investigate the convergence rate of optimizing $\tilde{V}^{\pi,\tau}(\rho)$ with the exact gradient descent and compare it with the vanilla policy gradient method. We study the smoothness and Łojasiewicz inequality for the function $\tilde{V}^{\pi,\tau}(\rho)$ to prove the convergence rate, similar to [26]. The detailed proofs of all following theorems are provided in the appendix.

**Theorem 1.** *In the tabular setting with softmax parameterized policy $\pi_\theta$, maximizing $\tilde{V}^{\pi,\tau}(\rho)$ using policy gradient with the learning rate $\eta = \frac{(1-\gamma)^3}{(8M+\tau(4+8\log A))}$, for all $t > 1$, we have*

$$\tilde{V}^{\pi_\tau^*,\tau}(\rho) - \tilde{V}^{\pi_{\theta_t},\tau}(\rho) \leq C \cdot e^{-C_\tau(t-1)} \cdot \frac{M + \tau \log A}{(1-\gamma)^2}$$

*where $\pi_\tau^*$ is the optimal policy maximizing $\tilde{V}^{\pi,\tau}(\rho)$, $M$ is the bound of the absolute value of $r(s,a) + \tau \log \beta(a|s)$, $A$ is the size of action space, $S$ is the size of state space, $C_\tau \propto \frac{(1-\gamma)^4}{(8M/\tau+4+8\log A) \cdot S}$, and $C$ is a constant independent with $t$ and $\tau$.*

Theorem 1 states that KL regularized expected return $\tilde{V}^{\pi,\tau}(\rho)$ can be optimized with a convergence rate $\mathcal{O}(e^{-t})$ rather than the $\mathcal{O}(1/t)$, the convergence rate of vanilla policy gradient for expected return alone. The faster convergence inspires us to optimize $\tilde{V}^{\pi,\tau}(\rho)$ to reach policy $\pi_\tau^*$, then use $\pi_\tau^*$ as initialization, gradually decrease the temperature $\tau$ towards 0, and eventually move from $\pi_\tau^*$ to $\pi^* = \arg\max_\pi V^\pi(\rho)$. With a reasonable value of $\tau$, we enjoy a linear convergence rate toward $\pi_\tau^*$ from the randomly initialized policy $\pi_\theta$. As $\tau$ decreases, $\pi_\tau^*$ gets closer to $\pi^*$. The final optimization of $V^{\pi_\theta}(\rho)$ from $\pi_\tau^*$ can be much faster than from a randomly initialized $\pi_\theta$.

We construct a toy example to illustrate this motivation. In the grid world (Fig. 1a), the start state, annotated with 'S', is in the center and the terminal states are marked in yellow. There are only two states with positive rewards (0.9 and 1). There are four actions {up, down, left, right}. A badly initialized policy $\pi_{\theta_0}$ is shown as arrows in Fig. 1a). The initialization results in a poor policy, having high tendency to go right toward a terminal state with zero reward. The vanilla policy gradient method (i.e. maximizing $V^\pi(\rho)$ with true gradient) starting from this initial point takes more than 7000 iterations to escape a sub-optimal solution (Fig. 1b). In contrast, we escape the sub-optimal solution much faster when applying the continuation method to update the policy with the gradients of $\tilde{V}^{\pi,\tau}(\rho)$, where the behavior policy $\beta(\cdot|s) = [u_1, u_2, u_3, u_4]$ with $u_i, i = 1, 2, 3, 4$ randomly sampled from $\mathcal{U}[0, 1]$ and

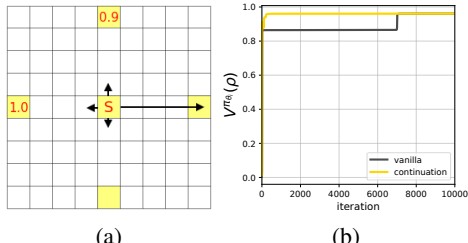

(a)          (b)

Figure 1: (a) A grid world with sparse rewards. (b) Learning curve of the value of learned policy $\pi_{\theta_i}$. We conduct a hyper-parameter search for the learning rate 5,1,0.5,0.1,0.05,0.01,0.005,0.001 and report the best performance for each method.

---

**Algorithm 1** Soft Policy Iteration through Continuation Method

---

1: Initialize: actor network $\pi_\theta$, ensemble critic network $\{Q_{\phi^{(1)}}, Q_{\phi^{(2)}}, \cdots, Q_{\phi^{(K)}}\}$, behavior policy network $\beta_\psi$, penalty coefficient $\tau$, decay rate $\lambda$, number of iterations $I$ for each $\tau$
2: Input: training dataset $D = \{(s_i, a_i, r_i, s_i')\}_{i=0}^N$
3: **for** update $j = 0, 1, \cdots$ **do**
4:     Sample batch of data $\{(s_i, a_i, r_i)\}_{i=1}^B$ from $D$
5:     *# Learn the behavior policy with behavior cloning objective*
6:     Update $\psi$ to maximize $\frac{1}{B} \sum_{i=1}^B \log \beta_\psi(a_i|s_i)$
7:     *# Train the critic network*
8:     Update $\phi^{(k)}$ to minimize the temporal difference $\frac{1}{B} \sum_{i=1}^B \left( r_i + \gamma V(s_i') - Q_{\phi^{(k)}}(s_i, a_i) \right)^2$
9:     where $V(s) = \frac{1}{K} \sum_{k=1}^K \mathbb{E}_{a \sim \pi_\theta(\cdot|s)}(Q_{\phi^{(k)}}(s, a)) - \tau KL(\pi_\theta(\cdot|s)|\beta_\psi(\cdot|s))$
10:    *# Train the actor network*
11:    Update $\theta$ to maximize $\frac{1}{B} \sum_{i=1}^B \left[ \frac{1}{K} \sum_{k=1}^K \mathbb{E}_{a \sim \pi_\theta(\cdot|s_i)}(Q_{\phi^{(k)}}(s_i, a)) - \tau KL(\pi_\theta(\cdot|s_i)|\beta_\psi(\cdot|s_i)) \right]$
12:    *# Decay the weight of KL regularization $\tau$ for every $I$ updates*
13:    **if** $j \mod I = 0$ **then**
14:       $\tau \leftarrow \tau * \lambda$
15:    **end if**
16: **end for**

---

normalized for each state $s$. In Fig. 1b, as we decrease $\tau$, the value of learned policy $\pi_{\theta_i}$ for each iteration $i$ quickly converges to the optimal value. In other words, optimizing a sequence of objective functions $\tilde{V}^{\pi,\tau}(\rho)$ can reach the optimal solution for $V^\pi(\rho)$ significantly faster.

### 3.2 SOFT POLICY ITERATION WITH KL REGULARIZATION

As explained in the previous section, we focus on the new objective function $\tilde{V}^{\pi,\tau}(\rho)$, which can be optimized more efficiently, and use continuation method to relax toward optimizing $V^\pi(\rho)$. Batch RL adds the complexity of estimating the gradient of $\tilde{V}^{\pi,\tau}(\rho)$ with respect to $\pi$ from a fixed set of trajectories. We propose to adapt soft actor-critic[16], a general algorithm to learn optimal maximum entropy policies in batch RL for our use case. We change the entropy regularization to KL regularization and derive the soft policy iteration to learn KL regularized optimal policy. For a policy $\pi$ and temperature $\tau$, the soft state value is defined in Eq. 1 and soft Q function is defined as:

$$\tilde{Q}^{\pi,\tau}(s,a) = r(s,a) + \gamma \mathbb{E}_{s' \sim \mathcal{P}(\cdot|s,a)} \tilde{V}^{\pi,\tau}(s') \tag{2}$$

In the step of soft policy evaluation, we aim to compute the value of policy $\pi$ according to the minimum KL divergence objective $\tilde{V}^{\pi,\tau}(\rho) = \mathbb{E}_{s \sim \rho}[\tilde{V}^{\pi,\tau}(s)]$. According to Lemma 1 in Appendix, the soft Q value can be computed by repeatedly applying the soft bellman backup operator.

$$\mathcal{T}^{\pi,\tau}Q(s,a) = r(s,a) + \gamma \mathbb{E}_{s' \sim \mathcal{P}(\cdot|s,a)}(V(s')), \text{ where } V(s) = \mathbb{E}_{a \sim \pi(\cdot|s)}\left[Q(s,a) - \tau \log \frac{\pi(a|s)}{\beta(a|s)}\right].$$

In the step of policy improvement, we maximize the expected return based on Q-value evaluation with the KL divergence regularization. The following policy update can be guaranteed to result in an improved policy in terms of its soft value (Lemma 2 in Appendix).

$$\pi_{new}(\cdot|s) = \arg\max_{\pi \in \Pi} \left[ \mathbb{E}_{a \sim \pi(\cdot|s)}\left(\tilde{Q}^{\pi_{old},\tau}(s,a)\right) - \tau KL(\pi(\cdot|s)|\beta(\cdot|s)) \right] \tag{3}$$

$$\text{where} \quad KL(\pi(\cdot|s)|\beta(\cdot|s)) = \mathbb{E}_{a \sim \pi(\cdot|s)}\left[\log \frac{\pi(a|s)}{\beta(a|s)}\right]$$

The soft policy iteration algorithm alternates between the soft policy evaluation and soft policy improvement, and it will provably converge to the optimal policy maximizing the objective $\tilde{V}^{\pi,\tau}(\rho)$.

**Theorem 2.** *Repeated application of soft policy evaluation and soft policy improvement converges to a policy $\pi_\tau^*$ such that $\tilde{Q}^{\pi_\tau^*,\tau}(s,a) \geq \tilde{Q}^{\pi,\tau}(s,a)$ for any $\pi \in \Pi$ and $(s,a) \in \mathcal{S} \times \mathcal{A}$.*

The soft policy iteration finds a policy $\pi_\tau^*$ with optimal soft Q value for each state-action pair and hence gets the optimal value of $\tilde{V}^{\pi,\tau}(\rho)$. Here we propose to use the soft policy iteration to solve objectives $\tilde{V}^{\pi,\tau}(\rho)$ with decreasing value of $\tau$ and move back to the objective $V^\pi(\rho)$ as $\tau = 0$. The method is guaranteed to asymptotically converge to the optimal policy $\pi^*$ for the objective $V^\pi(\rho)$.

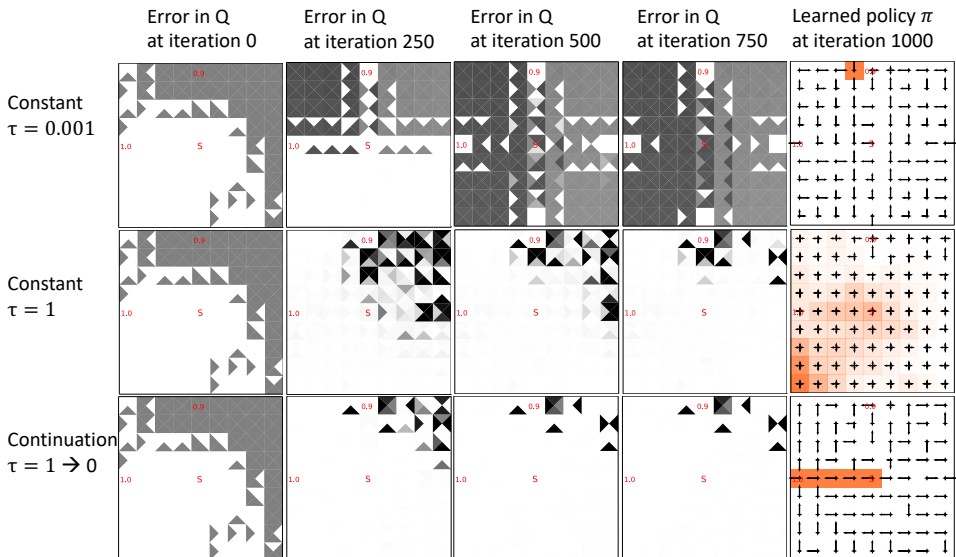

Figure 2: Visualization of the error in soft Q value estimation and quality of the learned policy. In the first four columns, triangles represent the error for actions that move in different directions. Darker color indicates higher error. To investigate the performance of the learned policy $\pi_{\theta_{1000}}$, the length of arrows represents the probability of taking each actions in each states. We run $\pi_{\theta_{1000}}$ in the grid world and visualize the visitation count in the last column (heatmap). Darker color means more visitation.

**Theorem 3.** *Let $\pi_\tau^*(a|s)$ be the optimal policy from soft policy iteration with fixed temperature $\tau$. We have $\pi_\tau^*(a|s) \propto \exp\left(\frac{\tilde{Q}^{\pi_\tau^*,\tau}(s,a)}{\tau}\right)\beta(a|s)$. As $\tau \to 0$, $\pi_\tau^*(a|s)$ will take the optimal action $a^*$ with optimal Q value for state s.*

## 3.3 ERROR IN VALUE ESTIMATE

In the previous section, we show that the soft policy iteration with the continuation method provably converges to the global optimal policy maximizing expected return. However, in batch RL with a fixed dataset and limited samples, we cannot perform the soft policy iteration with KL regularization in its exact form. Specifically, in the policy evaluation step, when the learned policy $\pi$ deviates for the behavior policy $\beta$, and chooses the state-action pair $(s, a)$ rarely visited by $\beta$, the estimation of target $r(s, a) + \gamma \mathbb{E}_{s' \sim \mathcal{P}(\cdot|s,a)}(V(s'))$ can be very noisy. The error in the value estimate $Q(s, a)$ will be further propagated to other state-action pairs through the bellman update. Finally, inaccurate value estimation will cause errors in the policy improvement step, resulting in a worse policy. On the other hand, if we constrain the learned policy $\pi$ to be very close to the behavior policy $\beta$, we can expect the policy evaluation to be reliable and safely update the learned policy. The tight constraint however prevents $\pi$ to be much better than $\beta$ due to the conservative update.

On the grid world, we study this problem of value estimation with different values of $\tau$. Figure 2 visualizes the propagation of Q value estimation errors and the learned policies. We assume a mediocre behavior policy tending to move left and down. For the rarely visited states in the upper right part of the grid, there are errors in the value estimation of $\tilde{Q}^{\pi,\tau}(s, a)$, i.e. $|Q(s, a) - \tilde{Q}^{\pi,\tau}(s, a)| > 0$ where $Q(s, a)$ is the Q value we learn during training and $\tilde{Q}^{\pi,\tau}(s, a)$ is the ground truth soft Q value. Because the bad initial policy (Fig. 1a) tends to move towards the right part, without a strong KL regularization, the policy evaluation can be problematic due to the errors of value estimation in the right part of the grid world. In Fig. 2, with a small KL regularization weight $\tau = 0.001$, the first row shows that errors even propagate to the frequently visited states by the behavior policy. On the other hand, when we set a large value of $\tau = 1$ (second row), the error $|Q(s, a) - \tilde{Q}^{\pi,\tau}(s, a)|$ is smaller. Yet the performance of the learned policy is not much better than the behavior policy. Our continuation method gradually moves the policy update between these two spectra. The value estimation benefits from the gradually relaxed KL regularization and the errors remain small. The last column of Fig. 2 visualizes the learned policy in these methods. With constant $\tau = 0.001$, the wrong value estimates in some states mislead the agent. It fails to visit any terminal state and gets stuck at the state in dark orange. With constant $\tau = 1$, the tight constraint of KL divergence makes the learned policy close to the behavior policy, mostly visiting the left bottom part of the environment. With continuation method, the agent learns to always take the optimal path moving left directly and obtains the highest expected return. More details of this example are provided in the appendix.

In the toy example, gradually relaxing KL regularization towards zero alleviates the propagation of errors in the soft Q estimate and helps the agent converge to the optimal policy. In more complicated domains, we find that as $\tau$ decays close to 0, the policy evaluation is still erroneous. To mitigate this issue, we introduce an ensemble of critic networks $\{Q_{\phi^{(1)}}, Q_{\phi^{(2)}}, \cdots, Q_{\phi^{(K)}}\}$ to approximate the soft Q value, and monitor the variance of value estimation in different critic networks to measure the uncertainty. Given a batch of data samples $\{s_i\}_{i=1}^B \subset \mathcal{D}$, $var(Q^\pi) = \frac{1}{B}\sum_{i=1}^B \mathbb{E}_{a\sim\pi(\cdot|s_i)}var(Q_{\phi^{(1)}}(s_i,a), Q_{\phi^{(2)}}(s_i,a), \cdots, Q_{\phi^{(k)}}(s_i,a))$ indicates whether the current policy $\pi$ tends to take actions with highly noisy value estimation.

Our method is summarized in Algorithm 1. Instead of running the soft policy evaluation and policy improvement until convergence, we alternate between optimizing the critic network and actor network with stochastic gradient descent. We set $\tau$ to large value initially and let the KL divergence term dominate the objective, thus performing behavior cloning. We record a moving average of the Q value estimation variance $var(Q^{\pi,\tau_0})$ over 1000 updates at the end of the phase. After that, we decay the temperature gradually with $\lambda = 0.9$ every $I$ steps. When the moving average of the Q value estimation variance $var(Q^{\pi,\tau})$ is large compared with the initial value $var(Q^{\pi,\tau_0})$ (i.e. at the end of behavior cloning), we no longer trust the value estimate under the current temperature $\tau$ and take the policy checkpointed before the temperature decays to this $\tau$ as our solution.

## 4 EXPERIMENTS

### 4.1 MUJOCO

We evaluate our method with several baselines on continuous control tasks. We train a Proximal Policy Optimization agent [28] with entropy regularization for 1000 million steps in the environments. We parameterize the policy using Gaussian policies where the mean is a linear function of the agent's state $\theta^T s$ and variance is an identity matrix to keep the policy simple as introduced in [3]. To generate training datasets $\mathcal{D}$ with varying quality, we construct the behavior policy by mixing the well-trained policy $\mathcal{N}(\theta_{opt}^T s, 0.5\mathbb{I})$, i.e. checkpoint with the highest score during training, and a poor policy $\mathcal{N}(\theta_0^T s, 0.5\mathbb{I})$ , i.e. checkpoint at the beginning of the training, with the weight $\alpha$. Then the behavior policy $\beta(\cdot|s)$ is $\mathcal{N}(((1-\alpha)\theta_{opt} + \alpha\theta_0)^T s, 0.5\mathbb{I})$. We generate trajectories and store a total of one million data samples from the mixed behavior for different values of the coefficient $\alpha$.

The architecture of the target policy is the same as the behavior policy. We consider six baseline approaches: BCQ [14], BEAR [21], ABM+SVG [29], CRR [33], CQL [22], BRAC [34]. For a fair comparison, the architectures of the ensemble critic network and the policy network are the same in the baselines and our method, except BCQ which has no policy network. To evaluate and compare the methods, we run the learned policy in the environments for 100 episodes and report the average episode reward in Fig 3. As for the continuation method, we report the score of policy checkpointed last with reasonable value estimation variance, as explained in Section 3.3. For the baselines, we report the score of the final policy when we terminate the training at 1.5M updates.

| | $\alpha$ | BCQ | BEAR | ABM | CRR | CQL | BRAC | Ours |
|---|---|---|---|---|---|---|---|---|
| | 0.2 | 1908.0 ±327.0 | 1661.2 ± 163.3 | 1814.2 ±176.9 | 1861.5 ±112.8 | 1962.6 ±194.1 | **2375.4 ± 181.3** | 2085.8 ±204.8 |
| Hopper | 0.4 | 876.0 ±462.2 | 1028.9 ± 49.0 | 1113.0 ±201.3 | 1281.0 ±218.2 | 1262.0±116.7 | 991.0 ± 39.5 | **1463.7 ±195.0** |
| | 0.6 | 429.9 ±198.1 | 898.6 ± 28.2 | 1402.6 ±350.5 | 791.1 ±87.0 | 609.1 ±35.1 | 573.7 ± 41.6 | **1524.3 ±511.4** |
| | 0.8 | **450.3 ±174.2** | 3.3 ± 4.3 | 3.6 ±0.0 | 304.2 ±36.3 | 295.7 ±41.9 | 83.6 ± 113.1 | 234.8 ±26.4 |
| | 0.2 | 1765.8 ±41.2 | 2143.4 ± 18.3 | **2168.3 ±26.1** | 2154.1 ±6.6 | 2105.7 ±37.2 | 1649.3 ± 64.3 | 2149.3 ±32.9 |
| Half | 0.4 | 1336.4 ±35.6 | 1809.3 ± 18.1 | **1914.7 ±13.2** | 1860.1 ±38.2 | 1811.9 ±60.7 | 1744.5 ± 15.1 | 1839.9 ±16.9 |
| Cheetah | 0.6 | 513.6 ±35.6 | 1149.9 ± 39.3 | 1457.9 ±36.4 | 1161.7 ±11.7 | 1156.6 ±43.3 | 1245.1 ± 67.2 | **1524.8±37.6** |
| | 0.8 | 179.0 ±14.1 | **700.6 ± 11.4** | 600.4 ±8.6 | 572.8 ±14.3 | 605.8 ±44.5 | 499.1 ± 11.0 | 595.4± 19.1 |
| | 0.2 | 1400.2 ±30.9 | 1414.6 ± 19.4 | 1405.2 ±57.3 | 1442.9 ±19.5 | 1385.2 ±73.4 | 437.5 ± 638.9 | **1467.3 ±45.2** |
| Walker | 0.4 | 965.9 ±69.0 | 1179.5 ± 21.2 | 1259.3 ±27.7 | 1233.5 ±35.8 | 979.6 ±83.5 | **1311.5 ± 29.7** | 1223.5±15.7 |
| | 0.6 | 266.3 ±56.9 | 486.3 ± 43.3 | 664.9 ±37.5 | 529.9 ±46.8 | 218.8 ±16.9 | 559.1 ± 37.6 | **872.9 ±228.7** |
| | 0.8 | 3.5 ±5.0 | **10.9 ± 0.9** | 5.9 ±0.6 | 10.7 ±0.3 | 3.3 ±6.5 | 5.6 ± 0.9 | 7.0± 6.3 |

Table 1: Results on Mujoco. We show the average and standard deviation of the scores in 5 independent runs.

Tab. 1 shows that our method outperforms all the baselines on 5 settings. On the dataset with relatively reasonable quality (i.e. $\alpha = 0.2, 0.4, 0.6$), ours performs comparable or better than the baselines. With $\alpha = 0.2$, i.e., close to optimal behavior policy, all the methods perform similarly and

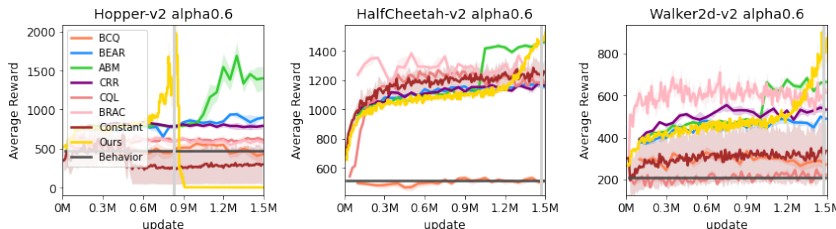

Figure 3: Learning curves of average reward over 5 runs on Mujoco tasks with $\alpha = 0.6$. The shaded area with light color indicates the standard deviation of the reward. The gray vertical lines on the yellow curves indicate where we take the checkpointed policy, according to the measure of Q value variance, as our final solution.

|       | Amidar | Asterix | Breakout | Enduro | MsPacman | Qbert | Seaquest | SpaceInvaders |
|-------|--------|---------|----------|--------|----------|-------|----------|---------------|
| BCQ   | 154.4  | 2466.7  | 203.8    | 604.7  | 2299.7   | 4088.3 | 4420.0  | 726.5         |
|       | ±11.0  | ±273.4  | ±19.6    | ±39.7  | ±150.2   | ±332.8 | ±548.7  | ±58.7         |
| REM   | 32.0   | 741.4   | 3.1      | 244.2  | 1997.4   | 2062.9 | 474.0   | 678.4         |
|       | ±3.5   | ±236.7  | ±0.0     | ±19.8  | ±6.4     | ±511.5 | ±61.9   | ±41.3         |
| CQL   | 145.0  | 2618.7  | **253.7**| 206.6  | 2234.6   | 4094.7 | 4652.6  | 493.5         |
|       | ±1.6   | ±102.1  | **±14.4**| ±5.8   | ±203.2   | ±74.1  | ±2017.1 | ±11.4         |
| Ours  | **174.5** | **3476.7** | 199.0 | **922.9** | **2494.0** | **4732.5** | **9935.0** | **1070.3** |
|       | **±7.1** | **±229.0** | ±32.0 | **±31.7** | **±301.3** | **±172.5** | **±1175.9** | **±137.1** |

Table 2: Results on Atari, the mean and standard deviation of scores achieved in 3 independent runs.

one can achieve a good return by simply cloning the behavior policy. With $\alpha = 0.8$, i.e., low-quality behavior policy, there are few good trajectories in the dataset for any methods to learn. The advantage of our method is most obvious when $\alpha = 0.6$ (Fig. 3), as the dataset contains trajectories of both high and low cumulative rewards. Our method can learn from the relatively large number of good trajectories and at the same time deviate from the behavior policy to avoid those bad trajectories and achieve higher rewards. In Fig. 3, 'Constant' is the method of optimizing KL regularized expected reward with constant value of $\tau$. We search several values of $\tau$ and report the best result. We can see that gradually relaxing constraint performs better than the fixed constraint. In Fig. 3(left), as $\tau$ decays to close to 0, the learned policy can degrade due to errors in the Q estimation, the stopping condition explained in Section 3.3 is however able to identify a good policy before the degenerating point. More experimental details are in the Appendix.

### 4.2 ATARI

We further study our method on several Atari games from the Arcade Learning Environment (ALE) [4]. The rich observation space requires more complicated policies and makes policy optimization even more challenging. We focus on eight games and generate the datasets as discussed in Fujimoto et al. [13]. We use a mediocre DQN agent, trained online for 10 million timesteps (40 million frames). The performance of the DQN agent is shown as 'Online DQN' in Fig. 4. We add exploratory noise on the DQN agent (at 10 million timesteps) to gather a new set of 10 million transitions, similar to [13]. The line "Behavior" in Fig. 4 shows the average of trajectory reward in the dataset $\mathcal{D}$. The dataset $\mathcal{D}$ is used to train each offline RL agent. We compare with BCQ [13], REM [2] and CQL [22] because they are recently proposed offline RL algorithms and work well on Atari domain. For evaluation, we run 10 episodes on the Atari games with the learned policies and record the average episode reward (Fig. 4). Tab. 2 summarizes the performance of BCQ, REM and CQL after 6M updates. For our method, we report the score before the variance of Q estimate becomes too high.

Our approach achieves higher scores than the baselines on 7 out of 8 games, and perform comparably on the other one. Agarwal et al. [2] reports that REM performs well in the dataset consisting of the entire replay experiences collected in the online training of the DQN agent for 50M timesteps (200M frames). We hypothesize that learning on the entire replay experience makes the setup easier as the training dataset contains more exploratory and higher quality trajectories. With the dataset of much smaller size and worse quality, REM performs poorly in this single behavioral policy setting. We use the same architecture of the critic network for both our method and BCQ with ensemble of 4 Q networks. As mentioned in [13], BCQ only matches the performance of the online DQN on most games. In contrast, ours is able to outperform online DQN significantly on several games. As presented in [22], on the Atari dataset, CQL performs better than REM, while our method outperforms CQL in 7 out of 8 datasets.

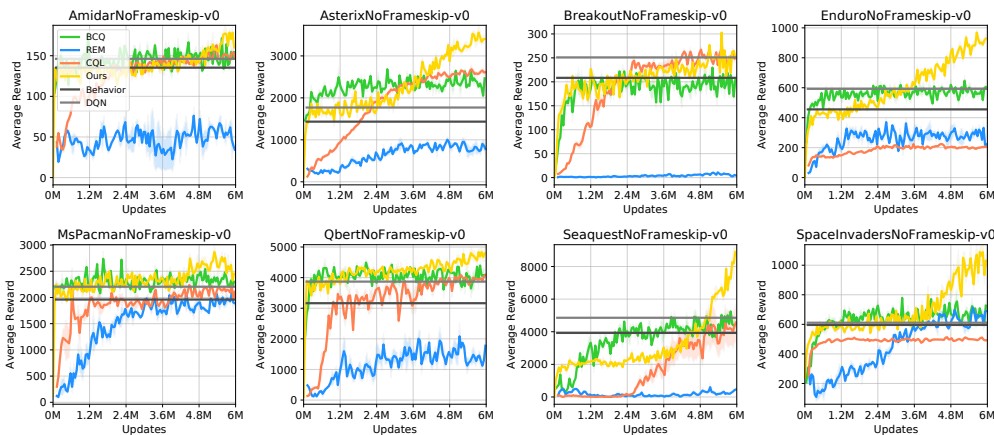

Figure 4: Learning curves of average reward over 3 runs on Atari games.

| | data with avg. reward 0.49 | | data with avg. reward 0.63 | | data avg. reward 0.81 | |
|---|---|---|---|---|---|---|
| sample size | 30,000 | 60,000 | 30,000 | 60,000 | 30,000 | 60,000 |
| Cross-Entropy | 74.3±0.01 | 73.5±0.01 | 82.0±0.01 | 81.3±0.01 | 86.7±0.00 | 85.6±0.00 |
| IPS | 80.5±0.03 | 82.1±0.02 | 86.4±0.02 | 88.0±0.01 | 87.2 ± 0.02 | 90.0 ± 0.01 |
| Ours | **83.0±0.01** | **85.8±0.00** | **88.7±0.01** | **90.1±0.00** | **89.4±0.00** | **90.5±0.00** |

Table 3: Results on MovieLens dataset. The number is precision at 10, i.e., the percent of recommended movies getting positive feedback when we use the well-trained policy to recommend top-10 movies to the test users. We report the mean and standard deviation of the precision over 20 independent runs,

## 4.3 RECOMMENDER

We also showcase our proposed method for building a softmax recommender agent. We use a publicly available dataset MovieLens-1M, a popular benchmark for recommender system. There are 1 million ratings of 3,900 movies (with the title and genre features) from 6,040 users (with demographic features). The problem of recommending movies for each user can be converted to a contextual bandit problem, where we aim to learn a target policy $\pi_\theta(a|s)$ selecting the proper action (movie) $a$ for each state (user) $s$ to get a high reward (rating) $r$ in a single step. The ratings of 5-score are converted to binary rewards using a cutoff of 4. To evaluate whether a learned target policy works well, ideally we should run the learned policy in real recommendation environments. However, such environments for online test are rarely publicly available. Thus, we use the online simulation method. We train a simulator to predict the immediate binary feedback from user and movie features, and the well-trained simulator can serve as a proxy of the real online environments, because it outputs the feedback for any user-movie pair. Similar to [25], we train the simulator with all records of logged feedback in MovieLens-1M dataset. The behavior policy is trained with partial data in MovieLens-1M. We then construct the bandit datasets $\mathcal{D} = \{s_i, a_i, r_i, \beta(a_i|s_i)\}_{i=1}^N$ of different size and quality, by using different behavior policies $\beta$ to select movies $a_i$ for users $s_i$ and getting the binary feedback $r_i$ from the well-trained simulator. We train offline RL agents on the generated dataset $\mathcal{D}$ and use the simulator to evaluate the learned policies on a held-out test set of users.

We compare our method with two baselines, as they are commonly used in current industrial recommender systems [10, 7]. (1) Cross-Entropy: a supervised learning method for the softmax recommender where the learning objective is the cross-entropy loss $J_{CE}(\theta) = -\frac{1}{N}\sum_{i=1}^N r_i \log \pi_\theta(a_i|s_i)$ (2) IPS: the off-policy policy gradient method introduced in [7] with the learning objective $J_{IPS}(\theta) = -\frac{1}{N}\sum_{i=1}^N \frac{sg(\pi_\theta(a_i|s_i))}{\beta(s_i,a_i)} r_i \log \pi_\theta(a_i|s_i)$ where $sg$ indicates a stop-gradient operation. $J_{IPS}(\theta)$ produces the same gradient as that of the function $-\frac{1}{N}\sum_{i=1}^N r_i \frac{\pi_\theta(a_i|s_i)}{\beta(a_i|s_i)}$. Thus, minimizing the loss $J_{IPS}(\theta)$ is to maximizing the expected return with importance sampling. (3) Ours: in the bandit setting, we simply perform IPS with gradually decaying KL regularization since estimating the soft Q from bellman update is not needed. Tab. 3 clearly demonstrates the advantage of our proposed method over the baselines. IPS can be viewed as vanilla policy gradient with importance sampling to correct the distribution shift. Our method clearly outperforms it across datasets collected using different behavior policies.

## 5    CONCLUSION

We propose a simple yet effective approach, soft policy iteration algorithm through continuation method to alleviate two challenges in policy optimization under batch reinforcement learning: (1) highly non-smooth objective function which is difficult to optimize (2) high variance in value estimates. We provide theoretical ground and visualization tools to help understand this technique. We demonstrate its efficacy on multiple complex tasks.

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
