# OpenReview forum: "Batch Reinforcement Learning Through Continuation Method"
_ICLR.cc/2021/Conference — ICLR 2021 Poster_

### Official Review · AnonReviewer3 · 2020-10-28
**Solid paper, but important novelty concerns**

**Rating:** 7
**Confidence:** 4

**Review:**

The authors propose a KL regularized approach for batch RL, where the importance of the KL term is reduced during learning. Theoretical guarantees are provided in the tabular domain. The algorithm is tested in several domains (MuJoCo, Atari, and a recommender task).

Strengths:
- Although the theoretical results are mostly straightforward extensions of existing results, they provide a solid backing to the method. In particular, Theorem 1 is an interesting result which motivates KL regularization in this setting.
- The experimental section is very thorough. I would prefer more seeds but the coverage over many domains and behavior policies is convincing evidence for the empirical success of the method.
- The appendix is very comprehensive. Although code is missing, the method and experiments are reproducible with the provided descriptions.
- The writing was clear, and the method was well-motivated. Overall, the paper feels fairly complete and well polished.

Weaknesses:
- There is one very glaring weakness to this paper- the proposed KL regularized approach for offline RL already exists (Wu et al., 2019) & related (Jacques, et al., 2019). This is additionally problematic as both methods are not cited or discussed in the paper. To the best of my knowledge, the continuation aspect is still novel as well as the theoretical contributions. However, a discussion on (Wu et al., 2019) is necessary.
- My first impression was that the variance solution to checkpointing $\tau$ in Section 3.3 was somewhat "hacky". On second thought, however, as suggested by the authors in the introduction, gradually reducing $\tau$ provides a mechanism for searching for the "optimal" value which trades between the constraint and learning on top of the proposed benefits of continuation for optimization. I think this is an interesting component of the method. On the plus side I think the checkpointing solves an important problem for batch RL, but on the downside, I think measuring the variance of the ensemble is not as well motivated as the rest of the method. I think the paper could benefit from additional discussion, or experiments which examine this aspect further.
- While the method is the "best" over a wide range of domains, the performance benefit seems fairly incremental. Consequently, for most users it's unclear if the benefits are sufficiently significant to warrant the additional complexity.

Recommendation:

So firstly, the novelty concerns absolutely need to be addressed and the mentioned papers cited/discussed in the paper. Regardless, I do feel like there is a meaningful contribution that builds on this prior work at both a theoretical and empirical level. As a result, I'm leaning on the side of accept.

References
- Wu, Yifan, et al. "Behavior regularized offline reinforcement learning." 2019.
- Jaques, Natasha, et al. "Way off-policy batch deep reinforcement learning of implicit human preferences in dialog." 2019.

**Post-Rebuttal

The authors have addressed most of my concerns. I have increased my score. Although the additional experiments on the variance/checkpointing are helpful I would still like to see more discussion in the paper itself.

---

> ### Author Response · Authors · 2020-11-23
> **Author Response to Reviewer 3**
>
> We thank R3 for all the comments. Please also refer to the common response above for the answers about BRAC and the measurement of variance.
>
> $\textbf{Q:}$  the performance benefit seems fairly incremental ... it's unclear if the benefits are sufficiently significant to warrant the additional complexity.
>
> $\textbf{A:}$ We respectfully disagree on this comment that our performance improvement is “fairly incremental”. On Atari datasets, the relative improvement over the best baseline (BCQ) is 37.6% on average, while the prior published work REM only achieves relative improvement 4.9% over the best baseline (calculated from Table 1 in [1]).
>
> |env                            |  Amidar   |   Asterix   |   Breakout  |  Enduro   |  MsPacman  |   Qbert   |  Seaquest   | SpaceInvaders |
> | -------- | -------- | --------| --------| --------| -------- | --------| --------| --------|
> |BCQ                            |  154.4     |   2466.7   |     203.8     |    604.7    |   2299.7        |  4088.3   |   4420.0     |      726.5           |
> |Ours                            |  174.5     |   3476.7   |     199.0     |    922.9    |   2494.0        |  4732.5   |   9935.0     |      1070.3         |
> |relative improvement  | 13.0%     |    40.9%   |     -2.4%    |    52.6%   |    8.4%          |   15.8%   |   124.8%    |      47.3%             |
>
> We apply the paired sample t-test to determine whether the true mean difference between our score and each baseline score on various datasets is greater than 0 for Mujoco and movie dataset.  All tests return p-value less than 0.05.
>
> |dataset | Mujoco | Mujoco | Mujoco | Mujoco | Mujoco | Mujoco |       Movie      |     Movie    |
> | -------- | -------- | --------| --------| --------| -------- | --------| --------| --------|
> |baseline |    BEAR   |    ABM    |    CRR    |    BCQ    |    CQL   |   BRAC   | Cross-Entropy |   IPS    |
> |p-value  |    0.0103  |  0.0279   |   0.0359 |    0.0026 |  0.0190  |  0.0112   |      0.0002        | 0.0003 |
>
> We would argue the added “complexity” of our method is minimal. The implementation is almost identical to SAC by simply replacing the entropy regularization with KL regularization and an additional step to decay the weight of the regularization. Our method can be implemented on top of SAC with less than 20 lines of code changes. The baselines, e.g., BEAR, CRR, ABM are essentially actor-critic with various regularization terms (e.g. MMD distance, KL divergence). Our method follows this general framework.  Regarding hyper-parameter search, while the baselines tune the weight of the regularization, we tune the initial KL regularization weight which is much less sensitive across different datasets. In conclusion, the complexity of our method in implementation and hyper-parameter search is similar to the baselines while our method performs favorably to the baselines.
>
> $\textbf{Q:}$ more seeds & missing code
>
> $\textbf{A:}$ We thank R3’s recognition that our experiment is convincing with coverage on various domains and behavior policies. We will add runs with more seeds later when the computational resources are available. We will also release the code with the camera-ready version.
>
> [1] Agarwal, R., Schuurmans, D., & Norouzi, M. (2020). An Optimistic Perspective on Offline Reinforcement Learning.

---

### Official Review · AnonReviewer4 · 2020-10-28
**Good paper, accept**

**Rating:** 9
**Confidence:** 4

**Review:**

Summary
-------------
The paper extends soft actor-critic (SAC) to the batch RL setting, replacing the policy entropy in the objective function with the KL divergence from the behavioral policy. The temperature parameter tau weighting the reward agains the KL term is annealed towards zero during the optimization process, which corresponds to starting with behavioral cloning for high values of tau and ending up with the standard reward maximization RL objective for tau=0. Theoretical analysis and experiments confirm the advantages of the proposed method.

Decision
-----------
I vote for accepting the paper. The idea of annealing the KL constraint is simple and elegant. Although it is very similar to other constrained policy update methods discussed in the Related Work section, the evaluation in the batch RL setting and demonstration of the improved convergence properties is novel. The execution is of high quality, with evaluations on tabular problems, MuJoCo, Atari, and a contextual bandit problem for movie recommendation.

Questions
--------------
1. As pointed out in Sec. 3.3, when the policy deviates too much from the behavioral policy, the value estimate becomes erroneous. Therefore, a criterion based on the ensemble variance of the Q-function estimates is proposed. Is there a way to derive such a criterion from first principles?
2. Can you relate your work to [1]?
3. Does your method work when the behavior policy is not known but only the dataset is available?
4. Can you quantify how far the optimal policy is allowed to be from the behavioral policy? For example, on a pendulum swing-up task, if the behavioral policy is taking random actions and the pendulum always jitters at the bottom, inferring the optimal policy from this data appears quite challenging. Can one give some criteria when the method is expected to work well?

References
---------------
[1] Nachum, O., & Dai, B. (2020). Reinforcement learning via fenchel-rockafellar duality. arXiv preprint arXiv:2001.01866.

---

> ### Author Response · Authors · 2020-11-23
> **Author Response to Reviewer 4**
>
> We thank R4 for all the comments. Please also refer to the common response above for the answers about the measurement of variance.
>
> $\textbf{Q:}$ Can you relate your work to [1]?
>
> $\textbf{A:}$ Yes, the optimization problem of maximizing the KL regularized objective function $\tilde{V}^{\pi, \tau}(\rho)$ can be converted to its Lagrangian duality form. Solving the dual optimization problem is connected to our policy evaluation and policy improvement steps. Therefore, the soft policy iteration algorithm can also be unified in the duality perspective proposed in [1]. The details of the derivation are provided in appendix E.
>
> $\textbf{Q:}$ Does your method work when the behavior policy is not known but only the dataset is available?
>
> $\textbf{A:}$ Yes, in all the experiments reported in the paper, the behavior policy is approximated via behavior cloning, as we stated in line 6 in Algorithm 1.
>
> $\textbf{Q:}$ Can you quantify how far the optimal policy is allowed to be from the behavioral policy? Can one give some criteria when the method is expected to work well?
>
> $\textbf{A:}$ We study the performance of our method for different behavior policy on the Mujoco dataset. The details are included in Appendix F. For each environment, we interpolate behavior policies between a well-trained policy learned on-policy ($\alpha=0$) and a random policy ($\alpha=1$). We find our method outperforms the behavior policy and various baselines in most cases, and the improvement is most significant with a mediocre behavior policy, that is when the data contains both trajectories of high and low cumulative rewards (i.e. $\alpha \in [0.4, 0.6]$). We plot the histogram of the trajectory cumulative reward in the behavior data and show the mean and the standard deviation of trajectory reward in each dataset in appendix F. We find the dispersion of the trajectory rewards, in particular the percentage of trajectories with reward better than one standard deviation of the mean (i.e. %1$\sigma$ trajectory), highly correlates with the improvement (i.e. the average trajectory reward we achieved minus the average trajectory reward of the behavior dataset).
>
> | env | Hopper | Hopper | Hopper |  Hopper | Cheetah| Cheetah | Cheetah|  Cheetah | Walker| Walker | Walker|  Walker |
> | -------- | -------- | --------| --------| --------| -------- | --------| --------| --------| -------- | --------| --------| --------|
> |$\alpha$       |     0.2    |     0.4    |     0.6    |     0.8    |     0.2    |     0.4    |     0.6    |     0.8    |     0.2    |     0.4    |     0.6    |     0.8    |
> |%$1\sigma$ trajectory |  10.2% |  16.7%  |    17%   |     8%   |   9.9%  |  12.5%  |  17.1%  |  14.4%  |     3%    |    24%   |    14 %  |    9%    |
> |behavior  | 2001 | 1167 |  470 |  246 |1724| 1327|  512 |  174 | 1382 |  917  |  210  |  0  |
> |ours         |    2097   |  1569 |   1648   |   226  | 2145 |  1824  | 1487  |   547  | 1441  |  1223 |   853  |   7    |
> |improvement |     96   |    402    |   1178   |   -20    |   421    |    497    |    975    |    373    |     59     |    306    |    643    |    7      |
>
>
> [1] Nachum, O., & Dai, B. (2020). Reinforcement learning via fenchel-rockafellar duality. arXiv preprint arXiv:2001.01866.

---

### Official Review · AnonReviewer1 · 2020-10-29
**New batch RL algorithm with natural intuition and solid empirical study**

**Rating:** 6
**Confidence:** 4

**Review:**

Summary of the paper:
This paper proposed a new batch RL algorithm based on the continuation method in numerical optimization. The proposed method makes use of KL regularization in the offline policy optimization process, with a decreasing temperature parameter -- which comes from the continuation method of optimization. The paper shows that the policy learning with the KL regularized value yields a faster convergence rate, as the motivation of using KL regularization, and a decreasing temperature will ensure the final convergence to the optimal policy. Experiments on Mojoco, atari, and a recommender data-set shows that the proposed algorithm is effective.

Justification for the score:
The proposed algorithm is a natural improvement over the current behavior regularized policy optimization methods. The intuition from the continuation method provides a justification for using a decreasing temperature. I think the main merit of this paper is the solid experiment, in simulation tasks with discrete actions and continuous actions, and in a real data-set. This contribution seems solid, but I have some concerns about Theorem 1 and important related work that is missed.

Detailed comments:
Pro:
1. The intuition behind the algorithmic change is clear enough. Section 3.1 gives the motivation of using a KL regularization and section 3.3 gives an illustrative example of why the continuation method can be better than a constant threshold.
2. The experiment section covers the standard RL benchmark in both continuous and discrete action settings. It additionally studied the performance of the proposed algorithm in a real dataset. Since while most batch RL work only run experiments in simulation tasks, this is a good step to bring the algorithm close to the motivation of doing batch RL.

Cons:
1. Theorem 1 seems to be a bit disconnected from the main contribution of the paper and it's hard to understand its role for me. Later theoretical analysis and practical approximation are all based on the policy iteration algorithm, but Theorem 1 seems to be based on policy gradient. Could the convergence rate improvement be shown in the policy iteration case? Additionally, Theorem 1 says "maximizing $\widetilde{V}^{\pi,\tau}$", does that mean theorem 1 consider the on-policy setting? In such a case what does $\beta$ mean?
2. The proposed algorithm seems to be very related to the BRAC framework and the algorithm BRAC with the KL value penalty. Why is it not mentioned at all? The BRAC paper may not directly use such a decreasing temperature or link it to the continuation method, but as prior work in batch RL, it also considered KL regularization and even studied using an adaptive regularization coefficient.
3. More recent batch RL algorithms like BRAC, CQL, etc also reported their result in mujoco and atari (CQL did) domains and seems to be better than BCQ, BEAR, and REM. An open dataset D4RL has the reported performance of those more recent baselines. I think in general it will be better to compare with these more recent baselines (especially BRAC since it's very related.).

---

> ### Author Response · Authors · 2020-11-23
> **Author Response to Reviewer 1**
>
> We thank R1 for all the comments. Please also refer to the common response above for the answers about the prior work BRAC.
>
> $\textbf{Q:}$ Theorem 1 seems to be a bit disconnected from the main contribution of the paper and it's hard to understand its role .... Could the convergence rate improvement be shown in the policy iteration case? … Does that mean theorem 1 consider the on-policy setting? In such a case what does β mean?
>
> $\textbf{A:}$ We would like to first acknowledge the gap between Theorem 1 and the Algorithm 1. Theorem 1 assumes tabular setting and access to exact/true gradient, while Algorithm 1 has to adapt to function approximation and offline setting with noisy gradient estimates. To the best of our knowledge, little has been proved on the convergence rate of policy gradient methods until very recently [1], which made the same assumption as ours. We prove that the soft policy iteration in Algorithm 1 optimizes the soft objective in Theorem 2 and 3, and leave the proof of its convergence rate as future work.  Several related works do suggest the possibility: [2] proves a precise equivalence between KL-regularized Q-learning and policy gradient; [3] utilizes visualization tools to show that in policy iteration, the loss landscape for the entropy regularized objective is promoting the global minimum while the vanilla objective function exerts a chaotic loss landscape
>
> Theorem 1 assumes access to the exact gradient with respect to the policy parameters. As a result the theorem does not necessarily imply an on-policy or off-policy setting.  $\beta$ can be interpreted as “prior policy” in on-policy settings [2][4] or “behavior policy” in off-policy settings.
>
>
> $\textbf{Q:}$ In general it will be better to compare with these more recent baselines (especially BRAC since it's very related.)
>
> $\textbf{A:}$ The original submission included many recent baselines, including ABM, CRR (which outperforms BEAR and BCQ on Mujoco), and discrete BCQ (which outperforms REM on Atari). Upon the reviewer’s suggestion, we add the comparison with BRAC (including both fixed regularization weight and adaptive regularization weight) and CQL[5] in experiment section 4.1.
>
> As observed in [5], CQL performs better than or comparable with BRAC and BEAR on Mujoco datasets. On datasets with behavior policy of reasonable quality (i.e. $\alpha=0.2, 0.4, 0.6$), ours performs comparable or better than the baselines. With $\alpha=0.2$, i.e., close to optimal behavior policy, all the methods perform similarly and one can achieve a good return by simply cloning the behavior policy. With $\alpha=0.8$, i.e., low-quality behavior policy, there are few good trajectories in the dataset for any methods to learn. The advantage of our method is most obvious when $\alpha=0.6$, as the dataset contains trajectories of both high and low cumulative rewards. Our method can learn from the good trajectories while at the same time deviate from the behavior policy to avoid those bad trajectories and achieve higher return.
>
> | env | Hopper | Hopper | Hopper |  Hopper | Cheetah| Cheetah | Cheetah|  Cheetah | Walker| Walker | Walker|  Walker |
> | -------- | -------- | --------| --------| --------| -------- | --------| --------| --------| -------- | --------| --------| --------|
> | $\alpha$|0.2|0.4|0.6|0.8|0.2 |0.4 |0.6|0.8 |0.2|0.4|0.6|0.8|
> |BRAC |523 | 2 | 460|185|**2249**|**1922**|1410|582| 573| 1079| 70 | 6|
> |CQL  |1963 |1262 | 609|**296**|2106|1812|1157 |**606**|1385|980|219|3|
> |Ours |**2097**|**1569**|**1648**|226|2145|1824|**1487**|547|**1441**|**1223**|**853**|**7**|
>
> In experiment section 4.2, we add the baseline CQL. As presented in [5], on the Atari dataset, CQL performs better than REM, and our method outperforms CQL in 7 out of 8 datasets.
>
> | env      |    Amidar   |    Asterix   |    Breakout   |    Enduro   |    MsPacman   |    Qbert   |    Seaquest   |    SpaceInvaders   |
> | -------- | -------- | --------| --------| --------| -------- | --------| --------| --------|
> |CQL   |145 | 2619 |**254**|207|2235|4095 | 4653| 494 |
> |Ours   |**175**|**3477**|199|**923**|**2494**|**4733**|**9935**|**1070**|
>
>
> [1] Jincheng Mei, Chenjun Xiao, Csaba Szepesvari, and Dale Schuurmans. On the global convergence rates of softmax policy gradient methods. arXiv preprint arXiv:2005.06392, 2020.
>
> [2] Schulman, J., Chen, X., & Abbeel, P. (2017). Equivalence between policy gradients and soft q-learning. arXiv preprint arXiv:1704.06440.
>
> [3] Bekci, R. Y., & Gümüş, M. (2020). Visualizing the Loss Landscape of Actor Critic Methods with Applications in Inventory Optimization. arXiv preprint arXiv:2009.02391.
>
> [4] Nachum, O., Dai, B., Kostrikov, I., Chow, Y., Li, L., & Schuurmans, D. (2019). Algaedice: Policy gradient from arbitrary experience. arXiv preprint arXiv:1912.02074.
>
> [5] Kumar, A., Zhou, A., Tucker, G., & Levine, S. (2020). Conservative Q-Learning for Offline Reinforcement Learning. arXiv preprint arXiv:2006.04779.

---

### Official Review · AnonReviewer2 · 2020-10-29
**A continuation-like method**

**Rating:** 4
**Confidence:** 5

**Review:**

While the title of the paper suggests that it leverages techniques from the vast literature on numerical continuation, the proposed approach is much more specific. The main idea consists in annealing the temperature parameter of the soft Bellman operator, and to warm start the the corresponding series of problems across time. In the language of numerical continuation, this approach fits under the umbrella of "natural parameter continuation", which I would describe succintly as "warm starting".

From the perspective of addressing the challenging optimization landscape presupposed in many problems, the combination of "continuation" + smooth approximation to the optimal equations makes sense. However, for me the narrative doesn't hold when the authors motivate their method in the context of offline batch policy gradient methods. As the authors point out, the main challenge associated with offline data is the inability to sample new data. This is problematic in the policy gradient setting because our derivative estimator only holds under the distribution under which the samples have been collected. As soon as the policy parameters are updated, the distributional shift should be addressed via an appropriate change of measure (or via a model). To me, this is the main challenge in the off-policy setting and the proposed continuation-based solution does not address this issue.  I view this as a Monte Carlo estimation problem first; not one pertaining to the optimization landscape.

Perhaps the paper should have been named differently because the remaining theoretical contributions in the paper do not pertain to "continuation" per se. Theorem 1 provides a bound on the policy gradient methods with a softmax policy (this is different from the "soft" optimality equations). Theorem 2 and 3 follow from the results in Rust (1994, 1996) and in econometrics where the smooth (soft) Bellman operator has been widely used. Theorem 2 follows from the perspective of policy iteration as an application Newton-Kantorovich to the smooth Bellman optimality equations. Theorem 3 follows from Dini's theorem where $\lim_{\tau \to 0} T^\star_\tau v = T^\star v$ where $T_\tau$ would be the smooth (soft) Bellman operator and $T^\star$ the usual Bellman operator. See Rust 1996 "Numerical Dynamic Programming in Economics", section 4, more specifically equations 4.2 and 4.4

---

> ### Author Response · Authors · 2020-11-17
> **Author Response to Reviewer 2 (1/2)**
>
> We thank R2 for all the comments. Here we reply to the questions raised in the review.
>
> $\textbf{Q:}$ From the perspective of addressing the challenging optimization landscape ..., ... "continuation" + smooth approximation ... makes sense. However, the narrative doesn't hold when the authors motivate their method in the context of offline batch policy gradient methods. ... the distributional shift should be addressed… this is the main challenge in the off-policy setting and the proposed continuation-based solution does not address this issue.
>
> $\textbf{A:}$ We respectfully disagree on this comment and address it in two stages. First, continuation helps the challenging optimization landscape of policy gradient methods even when exact/true gradient is available; second, continuation combined with the specific soft objective introduced addresses the distribution shift pertaining to offline batch policy gradient.
>
> As empirically constructed in [1] and proved in [2], the vanilla softmax policy gradient objective is extremely difficult to optimize even with the exact policy gradient. The expected return objective may exhibit suboptimal plateaus and exponentially many local optima in the worst case, and policy gradient on this objective converges to the global optimal policy at a rate of $\mathcal{O}(1/t)$. The entropy-regularized policy gradient on the other hand enjoys a significantly faster linear convergence rate $\mathcal{O}(e^{−t} )$. As explained in [2], introducing the entropy regularization bears similarity to adding a strongly convex regularizer in convex optimization, which is known to smooth the landscape and significantly improve the rate of convergence of first-order methods. A similar argument can be made in our case as we set a large temperature $\tau$ in equation (1) initially to smooth the optimization landscape. We prove the linear convergence rate of the soft objective defined in equation (1) in Theorem 1. The soft objective in (1) however converges to a different point than the optimal softmax policy optimizing the expected return, which is what we are interested in, we thus employ continuation to gradually decrease the temperature to solve the original problem.
>
> To address the distribution shift caused by offline data as pointed out by R2, we adapted the plain entropy regularization to the KL divergence to regularize toward the behavior policy. As a result, when we set a large temperature $\tau$ initially, we not only ease the optimization challenge but also limit the distribution shift by forcing the learned policy to be close to the behavior policy. As we gradually decrease the temperature, the learned policy is allowed to deviate from the behavior policy, and extrapolation errors can occur. The continuation, i.e., solving the series of approximated objectives as defined in equation (1) with warm-starting, however, helps us avoid roaming into regions with high extrapolation errors, as shown in Figure 2 and empirically demonstrated in the experiments. In summary, our proposed continuation method not only helps address “the challenging optimization landscape” but also alleviates the problem of extrapolation error induced by “the distributional shift” in offline RL.
>
> (to be continued)

---

> ### Author Response · Authors · 2020-11-17
> **Author Response to Reviewer 2 (2/2)**
>
> $\textbf{Q:}$ The remaining theoretical contributions in the paper do not pertain to "continuation" per se.
>
> $\textbf{A:}$ Our theoretical results are all aiming at supporting the use of continuation methods, or “natural parameter continuation” as commented by the reviewer for policy optimization, i.e., set up an easy problem, and “gradually transform back into the original problem and follow the solution as it moves from the solution of the easy problems to the solution of original problem” [3]. Theorem 1 proves that optimizing the KL regularized expected return is the easy problem, which can be solved more efficiently than the original problem. Theorem 2 and Theorem 3 proves that if we trace the solution of the easy problems, we will reach the solution of the original problem as the temperature $\tau$ goes to 0. We thank the reviewer for pointing us to the related literature. We will include them in the draft.
>
>
> $\textbf{Q:}$ Theorem 1 provides a bound on the policy gradient methods with a softmax policy, but this is different from the "soft" optimality equations.
>
> $\textbf{A:}$ We would like to first acknowledge the gap between Theorem 1 and Algorithm 1. Theorem 1 assumes tabular setting and access to exact/true gradient, while Algorithm 1 has to adapt to function approximation and offline setting with noisy gradient estimates. To the best of our knowledge, little has been proved on the convergence rate of policy gradient methods until very recently [2], which made the same assumption as ours. We prove that the soft policy iteration in Algorithm 1 optimizes the soft objective in Theorem 2 and 3, and leave the proof of its convergence rate as future work. Despite the gap, we would like to argue that Theorem 1 motivates the use of continuation and sheds light on the underlying mechanism of Algorithm 1, and our empirical results validate the efficacy of Algorithm 1. As the success of many algorithms beyond applications supported by theoretical proofs, for example, stochastic gradient descent (SGD) applied to deep neural networks, upper confidence bound (UCB) or Thompson Sampling beyond bandits settings, we hope our work can motivate future theoretical results in closing the gap.
>
> Thanks for reading! Please let us know if there are other comments we miss to address here.
>
> [1] Minmin Chen, Ramki Gummadi, Chris Harris, and Dale Schuurmans. Surrogate objectives for batch policy optimization in one-step decision making. In Advances in Neural Information Processing Systems, pp. 8825–8835, 2019.
>
> [2] Jincheng Mei, Chenjun Xiao, Csaba Szepesvari, and Dale Schuurmans. On the global convergence rates of softmax policy gradient methods. arXiv preprint arXiv:2005.06392, 2020.
>
> [3] Nocedal, J., & Wright, S. (2006). Numerical optimization. Springer Science & Business Media.

---

### Author Response · Authors · 2020-11-23
**Common Response and Summary of the Updates in the Paper (1/2)**

We thank all the reviewers for the constructive feedback. We are encouraged that the reviewers find our proposed method “well motivated” with “thorough experiments”, and the “theoretical results provide solid backing to the method”. We address the common questions here and will reply to each reviewer separately for their specific comments.

**Updates in the paper:**

- Section 2: Discussion of related work BRAC [1] and [2].
- Section 4.1: Comparison with baseline BRAC and CQL
- Section 4.2: Comparison with baseline CQL
- Appendix C: Experimental details of the baseline BRAC and CQL
- Appendix D: Study of variance of ensemble critic networks
- Appendix E: Study to relate our method with duality in RL algorithm
- Appendix F: Study of performance on datasets of different quality

**Discussion of related work BRAC (R1, R3):**

We thank the reviewers for pointing out the prior works. We added the discussion of BRAC [1] and [2] in the related work section in the revision. These prior works constrained the KL divergence between the target policy and the behavior policy through a fixed regularization weight or a fixed threshold. The fixed weight variant of BRAC is equivalent to our baseline method denoted as “Constant” in Figure 2&3, with some minor differences in implementation details (e.g., regularization in network parameters, fixing behavior policy after 30K updates).  Our main contribution on the other hand lies in introducing continuation to anneal the weight of the KL regularization for 1) solving the original problem of maximizing expected return to the best extent possible while addressing the distribution shift under batch RL; 2) more efficient optimization. We provided theoretical justification for introducing KL regularization and continuation for batch RL, a practical algorithm (Algorithm 1), and extensive experiments to showcase its efficacy. We would therefore argue the existence of these prior works does not weaken the novelty or contribution of our work.

We added the experimental comparison with BRAC using the implementation from [1]  in section 4.1.  As explained in section 1, the strength of the fixed constraint is a critical hyper-parameter that is hard to tune. We conduct the hyper-parameter search for BRAC as done in [1] (more details can be found in Appendix C), and find its performance to vary significantly over different environments. BRAC performs significantly worse on Hopper and Walker (as shown in the table below), similar to what we observed for the “Constant” baseline. The “best” hyper-parameters identified for these datasets often lead to erroneous updates with degenerating performance. This comparison further validates the merits of continuation method that gradually relaxing the constraint vs keeping a fixed constraint.

| env | Hopper | Hopper | Hopper |  Hopper | Cheetah| Cheetah | Cheetah|  Cheetah | Walker| Walker | Walker|  Walker |
| -------- | -------- | --------| --------| --------| -------- | --------| --------| --------| -------- | --------| --------| --------|
| $\alpha$ |  0.2 | 0.4 | 0.6 | 0.8 | 0.2 | 0.4 | 0.6 | 0.8 | 0.2 | 0.4 | 0.6 | 0.8 |
| BRAC |   523     |      2      |     460   |    185    |**2249**|**1922**|   1410   |**582**|    573    |  1079    |    70      |      6      |
| Ours   |**2097**|**1569**|**1648**|**226**|   2145   |   1824   |**1487**|    547    |**1441**|**1223**|**853**|**7**|

(to be continued)

---

### Author Response · Authors · 2020-11-23
**Common Response and Summary of the Updates in the Paper (2/2)**

**Variance of value estimation in ensemble Q networks(R3, R4):**

High Q estimation errors can mislead the policy updates. In our algorithm, we use the variance in the ensemble Q networks to detect high Q estimation error, preventing further annealing $\tau$. The Q estimation error can be decomposed into two parts: the bias of the Q network and the variance of Q estimations. In our submission, we use variance in the Q ensemble to approximate the Q estimation variance. We conducted a separate set of experiments by training the critic networks with jackknife resampling to obtain the estimation variance and showed these two setups lead to very similar results.

- **train different critic networks with the same data**

As in Algorithm 1 line 8, the critic networks in the ensemble are updated with the same batch of data, the discrepancy in the value estimation comes from the different initializations. On state-action pairs frequently visited by the behavior policy, the impact of initialization diminishes as training goes on, resulting in small variance. On state-action pairs less frequently, the large variance in the Q estimation persists. Similar to [3], the discrepancy among different networks is a proxy for the inverse visitation count of the state-action pair.

We empirically validate that the variance in the Q ensemble is a good proxy for the Q estimation error in Appendix D to motivate its usage as the stopping criterion. For each state-action pair in the grid world environment, we visualize the variance in the Q ensemble $var(Q_{\phi^{(1)}}(s,a), Q_{\phi^{(2)}}(s,a), \cdots, Q_{\phi^{(K)}}(s,a))$ and the error in Q estimation $\frac{1}{K}\sum_{k=1}^K Q_{\phi^{(k)}}(s,a) - \tilde{Q}^{\pi, \tau}(s,a)$, and show these two quantities are highly correlated. If the learned policy $\pi$ tends to select the state-action pairs with high variance in Q estimation, the error in the Q estimation on these state-action pairs will propagate to the other state-action pairs and mislead the policy update. We thus measure $E_{a\sim\pi} [var(Q_{\phi^{(1)}}(s,a), Q_{\phi^{(2)}}(s,a), \cdots, Q_{\phi^{(K)}}(s,a))]$ as the criteria to stop further annealing $\tau$.

- **train different critic networks with different data**

A more standard approach to estimate the variance is through data perturbation. We added experiments with training the critic network with delete-d jackknife resampling. To train each of four critic networks, we leave out $\frac{1}{4}$ data samples in the dataset (Note that the target value in the Q update is still averaged across the four critics. We found it to be important to the performance of the learned policy).
As shown below, the empirical results with critics trained on different data are similar to the ones we reported in the submission for Mujoco and Atari. The variances measured in these two approaches are quite similar as well.

 | env | Hopper | Hopper | Hopper |  Hopper | Cheetah| Cheetah | Cheetah|  Cheetah | Walker| Walker | Walker|  Walker |
| -------- | -------- | --------| --------| --------| -------- | --------| --------| --------| -------- | --------| --------| --------|
| $\alpha$ |  0.2 | 0.4 | 0.6 | 0.8 | 0.2 | 0.4 | 0.6 | 0.8 | 0.2 | 0.4 | 0.6 | 0.8 |
| Same |  2097 | 1569 | 1648 | 226 | 2145 | 1824 | 1487 | 547 | 1441 | 1223 | 853 | 7|
| Jackknife| 2072  | 1718 | 1404 | 230 | 2206  | 1840 | 1529 | 589 | 1462 | 1078 | 920 | 8 |

| env |    Amidar   |    Asterix   |    Breakout   |    Enduro   |    MsPacman   |    Qbert   |    Seaquest   |    SpaceInvaders   |
| -------- | -------- | --------| --------| --------| -------- | --------| --------| --------|
|Same        |      175      |     3477     |       199         |      923      |         2494        |     4733   |        9935      |            1070            |
|Jackknife   |      171      |     3890     |       217        |       908      |         2305        |     5389   |       9636      |           914              |


[1] Wu, Y., Tucker, G., & Nachum, O. (2019). Behavior regularized offline reinforcement learning. arXiv preprint arXiv:1911.11361.

[2] Jaques, N., Ghandeharioun, A., Shen, J. H., Ferguson, C., Lapedriza, A., Jones, N., ... & Picard, R. (2019). Way off-policy batch deep reinforcement learning of implicit human preferences in dialog. arXiv preprint arXiv:1907.00456.

[3] Burda, Y., Edwards, H., Storkey, A., & Klimov, O. (2018). Exploration by random network distillation. arXiv preprint arXiv:1810.12894.

---

### Decision · Program_Chairs · 2021-01-07
**Final Decision**

**Decision:**

Accept (Poster)

**Comment:**

The paper got a quite high disagreement in the scores from the reviewers. R2 voted for rejecting the paper as he did not see the connection of the algorithm to the continuation method and also that the continuation method does not address the distributional shift, which is one of the main problems for offlline RL. Yet, these concerns have been properly answered in the rebuttal of the authors and the distributional shift is also addressed by the continuation method by reducing the error in policy evaluation. Further concerns from the reviewers were raised in terms of related work to a similar algorithm (BRAC), which is also addressed in the revision of the paper.

The reviewers also identified the following strong points of the paper:
- The algorithm is a simple and very effective adaptation to SAC
- The presented results are exhaustive and convincing
- The paper provides strong theoretical results for the presented algorithm
- The authors did a very good job with their revision, adding more comparisons and ablation studies.

I agree that this paper very interesting and recommend acceptance.